# Homophilic Interaction of CD147 Promotes IL-6-Mediated Cholangiocarcinoma Invasion via the NF-κB-Dependent Pathway

**DOI:** 10.3390/ijms222413496

**Published:** 2021-12-16

**Authors:** Paweena Dana, Ryusho Kariya, Worachart Lert-itthiporn, Wunchana Seubwai, Saowaluk Saisomboon, Chaisiri Wongkham, Seiji Okada, Sopit Wongkham, Kulthida Vaeteewoottacharn

**Affiliations:** 1Department of Biochemistry, Faculty of Medicine, Khon Kaen University, Khon Kaen 40002, Thailand; paweena.dana@gmail.com (P.D.); woracle@kku.ac.th (W.L.-i.); saowaluksai@kkumail.com (S.S.); chaisiri@kku.ac.th (C.W.); sopit@kku.ac.th (S.W.); 2Cholangiocarcinoma Research Institute, Khon Kaen University, Khon Kaen 40002, Thailand; wunchana@kku.ac.th; 3Division of Hematopoiesis, Joint Research Center for Human Retrovirus Infection and Graduate School of Medical Sciences, Kumamoto University, Kumamoto 860-0811, Japan; ryushokariya@gmail.com (R.K.); okadas@kumamoto-u.ac.jp (S.O.); 4Department of Forensic Medicine, Faculty of Medicine, Khon Kaen University, Khon Kaen 40002, Thailand

**Keywords:** cholangiocarcinoma (CCA), cluster of differentiation 147 (CD147), extracellular matrix metalloproteinase inducer (EMMPRIN), proinflammatory cytokines, cancer invasion

## Abstract

Cholangiocarcinoma (CCA), an aggressive cancer of bile ducts, is a well-known chronic inflammation-related disease. The major impediment in CCA treatment is limited treatment options for advanced disease; hence, an alternative is urgently required. The role of CD147 on cytokine production has been observed in inflammation-related diseases, but not in CCA. Therefore, this study was focused on CD147-promoting proinflammatory cytokine production and functions. Proinflammatory cytokine profiles were compared between CD147 expressing CCA cells and CD147 knockout cells (CD147 KO). Three cytokines, namely interleukin (IL)-6, IL-8, and granulocyte–monocyte colony-stimulating factor (GM-CSF), were dramatically diminished in CD147 KO clones. The involvement of the CD147-related cytokines in CCA invasion was established. CD147-promoted IL-6, IL-8, and GM-CSF secretions were regulated by NF-κB nuclear translocation, Akt activation, and p38 phosphorylation. CD147-fostering IL-6 production was dependent on soluble CD147, CD147 homophilic interaction, and NF-κB function. The overexpression of specific genes in CCA tissues compared to normal counterparts emphasized the clinical importance of these molecules. Altogether, CD147-potentiated proinflammatory cytokine production leading to CCA cell invasion is shown for the first time in the current study. This suggests that modulation of CD147-related inflammation might be a promising choice for advanced CCA treatment.

## 1. Introduction

Cholangiocarcinoma (CCA) is an aggressive cancer of bile ducts, which is highly prevalent in the Northeast of Thailand [1]. It is a significant cancer-related health problem in the area. The major contributing factor of high CCA incidence in Thailand is the infection of the oncogenic liver fluke, *Opisthorchis viverrini* [1]. Silent progression, lack of pathognomonic signs and symptoms, and no accurate diagnostic tools lead to delayed diagnosis and metastatic progression [2]. Most patients are diagnosed with the advanced stages of disease, where curative therapy is not relevant, and only palliative treatment can be offered [3]. Hence, an urgent requirement of the treatment option for CCA patients is desired, particularly for patients with advanced disease.

The tight association between CCA and chronic inflammation is well recognized. Proinflammatory cytokines potentiate chronic inflammation-related CCA pathogenesis, and progression is established [4,5,6,7,8]. This process is independent of CCA causative agents. Increased proinflammatory cytokines, e.g., interleukin-6 (IL-6), IL-8, and transforming growth factor-beta (TGF-**β**), are currently in focus [4,5,9]. The contributions of these cytokines to precancerous lesion formation and stem cell transformation have been proposed [5,9]. In the CCA milieu, several sources of proinflammatory cytokines are proposed, including transformed cholangiocytes [4], cancer-associated fibroblasts [10], and cancer-associated immune cells [11].

A cluster of differentiation 147 (CD147) is a glycoprotein that was first discovered as a matrix metalloproteinase (MMP) inducer, so it is alternatively called extracellular MMP inducer (EMMPRIN) [12]. Pleiotropic effects of CD147 have, however, been demonstrated [13]. CD147 could induce proinflammatory cytokine production, causing the progression of inflammatory diseases, e.g., rheumatoid arthritis and atherosclerosis. Numbers of proinflammatory cytokines, such as IL-6, IL-8, and tumor necrosis factor-alpha (TNF-α), are involved with CD147 functions [14,15]. In head and neck cancer, the inhibition of CD147 alleviates proinflammatory cytokines, e.g., IL-1β, IL-6, and IL-8, mediated cancer cell proliferation in vitro and in vivo [16]. In chronic inflammation-related CCA, the overexpression of CD147 and its association with metastatic progression and metabolic alteration have been reported [17,18]. The involvement of CD147 and proinflammatory cytokine production, however, has never been explored. Therefore, the proinflammatory cytokine-promoting action of CD147 and its participation in CCA invasion were focused on in this study.

## 2. Results

### 2.1. Expressions of CD147 in CCA Cell Lines and CD147 Knockout Clones (CD 147 KO)

Both membranous and soluble CD147s are identified [19], and the activations of CD147s are modulated through either cyclophilin A (CyA)-dependent or CyA-independent manners [20]. Autocrine and paracrine homophilic interactions of CD147 were revealed [20,21,22]. Accordingly, CD147 expression was determined in three CCA cell lines, namely KKU-055, KKU-100, and KKU-213A. Total and soluble CD147s were detected in total cell lysate and conditioned media (CM) (Figure 1A,B). The results demonstrated that KKU-055 and KKU-100 had low levels of highly glycosylated CD147 (HG), while KKU-213A expressed a higher level of CD147 HG. The lowly glycosylated form (LG) was prominently observed in KKU-100 (Figure 1A). Soluble CD147 HG was observed only in KKU-213A CM (Figure 1B). To study the functional significance of CD147, CD147 KO#1 and #2 were generated and described as mentioned elsewhere [18]. The completely diminished CD147s in CD147 KO#1 and #2 are demonstrated in Figure 1C.

### 2.2. Depletion of CD147-Alleviated Proinflammatory Cytokine Expressions in CCA CM

To assess the association between CD147 and inflammatory cytokine production in CCA cells, the cytokine levels in KKU-213A and CD147 KO#1 CM were compared using human inflammatory cytokine arrays. The results are shown in Figure 2A. Relative levels of 20 cytokines, which were expressed lower than 0.8-fold or higher than 1.2-fold in CD147 KO#1, are presented in Figure 2B. Four cytokines, namely granulocyte–monocyte colony-stimulating factor (GM-CSF), interleukin (IL)-6, IL-8, and chemokine (C-C motif) ligand 2 (CCL2), were selected. GM-CSF, IL-6, and IL-8 were decreased to 0.15-, 0.15-, and 0.42-fold, while CCL2 was increased to 3.44-fold compared to those of parental cells. The array map is presented in Figure 2C.

To confirm the effect of CD147 on cytokine production, the levels of selected cytokines in CM of KKU-213A, CD147 KO#1, and CD147 KO#2 were determined by ELISA. The results demonstrated that CM from both CD147 KO clones contained lower IL-6, IL-8, and GM-CSF than those of parental cells (Figure 3A,C). CCL2 expressions, however, were inconclusive (Figure 3D) and were excluded from subsequent studies.

### 2.3. CD147-Related Cytokine-Potentiated CCA Cell Invasion

CD147-promoted CCA migration and invasion were previously reported by the current group [17,18], and the functions of CD147-potentiated proinflammatory cytokine production were established [14,15]. Moreover, the actions of specific proinflammatory cytokines, e.g., interferon (IFN)-γ, IL-1β, IL-6, and TNF-α, promoting CCA migration and invasion were previously demonstrated [23,24]. Little is known, however, regarding the involvement of CD147 in proinflammatory cytokine-mediated CCA invasion. Therefore, the involvement of IL-6, IL-8, and GM-CSF on CCA invasion was explored in CD147 KO#1 and low CD147-expressing KKU-055 cells. The results showed that IL-6, IL-8, and GM-CSF promoted CCA invasion (Figure 4A,B). IL-6 could promote the invasion of CD147 KO#1 to 3.40 ± 1.11-fold and those of KKU-055 to 4.15 ± 1.16-fold. IL-8 increased CD147 KO#1 and KKU-055 cell invasions to 3.88 ± 0.98 and 3.31 ± 1.07-fold, and GM-CSF could augment those cell invasions to 8.12 ± 2.10 and 2.88 ± 1.08-fold. The expressions of corresponding cytokine receptors were determined by RT-PCR, as demonstrated in Appendix A. The expressions of *IL-6R* and *CXCR1* (*IL-8R*) were detected in KKU-213A and CD147 KO#1 but were barely detectable in KKU-055. The expressions of GM-CSFRα were mainly observed in KKU-055 and CD147 KO#1, while it was hardly detected in KKU-213A. The expression of *CXCR2* (*IL-8R*) was measurable in all three cell lines.

### 2.4. Soluble CD147-Mediated Cell Invasion and IL-6 Production

Several molecular partners of CD147 have been revealed. These include homodimerization and transactivation of membranous CD147 by soluble CD147 [22]. The functional significance of soluble CD147-derived cancer cells and exogenous recombinant human CD147 (rhCD147) promoted cell invasion have been previously established in breast cancer [25,26]. Both full-length and the distal IgG-like domains of CD147 could interact with the membrane CD147 and mediate cell invasion [25,27]. These effects required the involvement of fibroblast-derived matrix metalloproteinase (MMP)-2 or cancer-derived membrane-type MMP [25,27]. Moreover, soluble CD147 could increase CD3^+^ CD25^+^ T cells in psoriatic patients [28] and induce IL-18 expression in an atherosclerotic model [29]. Therefore, the functions of soluble CD147 on CCA cell invasion and proinflammatory cytokine production were explored.

To elucidate the actions of soluble CD147 on CCA cell invasion and cytokine production, rhCD147 was used as the representative of soluble CD147. KKU-055 (low CD147-expressing cells), KKU-213A (high CD147-expressing cells), and CD147 KO#1 (CD147-null cells) were treated with rhCD147, and cell invasions were determined. The results showed that rhCD147 induced CCA cell invasion in KKU-055 and KKU-213A, but not in CD147 KO#1 (Figure 5A,B). Recombinant hCD147 could promote KKU-055 and KKU-213A invasions to 3.38 ± 0.77 and 2.51 ± 0.16-fold, but the same treatment could not increase the invasion of CD147 KO#1 cells (1.10 ± 0.12-fold).

The production of selected CD147-related proinflammatory cytokines was determined in rhCD147-treated cells. The results revealed that rhCD147 could induce IL-6, but not IL-8 and GM-CSF production (Figure 6A,C). Recombinant hCD147 significantly induced IL-6 production in KKU-055 (*p* < 0.05), but the elevation in KKU-213A did not reach statistical significance (*p* = 0.172). Treatment of rhCD147, however, could not stimulate IL-6 production in CD147 KO#1 (Figure 6A). Similar effects of soluble CD147 were not observed on IL-8 or GM-CSF production (Figure 6B,C). These results suggested the requirement of endogenous CD147 on soluble CD147-mediated CCA cell invasion and IL-6 production.

### 2.5. NF-kB, p38, and Akt Involved in CD147-Promoted Proinflammatory Cytokine Production

Several regulators are involved in IL-6, IL-8, and GM-CSF production, though only Akt, NF-κB, and p38 are regulated by CD147 (Figure 7A) [30]. Hence, these molecules might be necessary for CD147-mediated proinflammatory cytokine production. Decreased nuclear NF-κB, p65, in CD147 KO clones was previously demonstrated [18]. Expressions of p-Akt, Akt, p-p38, and p38 were assessed, and the results showed reductions in p-Akt and p-p38 (Figure 7B). The alteration of p38 in CD147 KO clones was inconclusive because p38 was reduced in CD147 KO#2, but not in CD147 KO#1.

To determine the involvement of Akt, NF-κB, and p38 on the production of selected proinflammatory cytokines, KKU-213A was treated with specific inhibitors, NF-κB nuclear translocation inhibitor-DHMEQ, Akt inhibitor-MK-2206, and p38 inhibitor-SB202190, for 24 h, and the levels of IL-6, IL-8, and GM-CSF in CM were determined. The results showed that DHMEQ effectively suppressed IL-6, IL-8, and GM-CSF (Figure 7C,E). MK-2206 and SB202190 could repress IL-6, but not IL-8 and GM-CSF. Nonetheless, dose-dependent suppression was not observed. The results suggested that NF-κB might be a key molecule in CD147-mediated cytokine production in CCA.

### 2.6. Inductions of BSG, IL-6, RELA, AKT1, and MAPK14 Observed in Clinical Samples

Diminished IL-6, IL-8, and GM-CSF were observed in CD147 KO clones. These were related to lower nuclear NF-κB, p-Akt, and p-p38 in CD147-null cells, and the involvement of Akt, NF-κB, and p38 on proinflammatory cytokine production, particularly IL-6, was confirmed by specific inhibitor treatment. Therefore, the expressions and correlations of these molecules in clinical samples of CCA were of interest. Gene expression profiles of *BSG* (CD147), *IL-6*, *RELA* (p65), *AKT1* (Akt), and *MAPK14* (p38) in CCA tissues were retrieved from three different datasets, including TCGA-CHOL (Dataset #1), GSE26566 (Dataset #2), and GSE89749 (Dataset #3). The expression levels of certain genes in CCA tissues (C) were compared with adjacent noncancerous tissues (A) or normal bile duct (N) obtained from the same dataset. The results showed the expressions of *BSG*, *IL-6*, *RELA*, *AKT1*, and *MAPK14* were relatively higher in CCA tissues when compared to normal counterparts (Figure 8A,E), except for the *IL-6* level in Dataset #3 (Figure 8B). Some comparisons did not reach statistical significance due to the limited number of the samples and the normal counterpart, especially in Dataset #3, which contained only two normal tissues. Moreover, when the correlations with gene expression were investigated, there was no correspondence between different datasets (Appendix A).

## 3. Discussion

CCA, an aggressive cancer of bile ducts, is a leading health problem in Thailand [1]. The majority of cases in the area are diagnosed at the late stage [3], for which only palliative treatment can be offered [31]. CCA is rare worldwide, but the incidence is currently rising [32]. Moreover, the tight relationship between chronic inflammation and CCA progression suggests targeting chronic inflammation might be a novel target for CCA treatment, particularly in advanced stages [4,5,6,7,8]. CD147 or EMMPRIN is a multifunctional glycoprotein related to inflammation-promoting disease progression and cancers [14,15,16]. The functional significance of CD147-promoted CCA metastasis was previously demonstrated by the present group [17,18]. CCA is an inflammation-related cancer; however, the roles of CD147 in proinflammatory cytokine production in CCA have never been explored. This led to the investigation of the participation of CD147 in proinflammatory cytokine production.

The expressions of membranous and soluble CD147 were observed in CCA cell lines. The connection of CD147 and proinflammatory cytokine production in the CCA cells was pursued by using CD147 KO cells. Comparing the proinflammatory cytokine profiles between the CD147-competent and CD147-compromised cells identified IL-6, IL-8, and GM-CSF as the candidate cytokines in CCA. The importance of these CD147-related cytokines was demonstrated by their invasion-promoting actions. This action seemed to be independent of endogenous CD147 because the cytokines could stimulate CCA invasions in both KKU-055, low CD147-expressing cells, and CD147 KO#1, CD-147 null cells. Moreover, an action of soluble CD147 was revealed by rhCD147. The rhCD147-promoted CCA cell invasion was CD147 dependent and detected only in CD147-expressing cells, KKU-055 and KKU-213A. The proinflammatory cytokine-promoting action of soluble CD147, however, could only be demonstrated in the production of IL-6, but not in IL-8 or GM-CSF. Declining nuclear NF-κB, active Akt (p-Akt), and active p38 (p-p38) in CD147 KO clones suggested that these molecules might be involved in CD147-promoting cytokine production. Only DHMEQ, NF-κB nuclear translocation inhibitor, could suppress the production of all three cytokines. This suggests a key function of NF-κB in CD147-modulated proinflammatory cytokine production. Akt and p38 inhibitors, namely MK-2206 and SB202190, could only reduce IL-6 production, not IL-8 and GM-CSF, implying complex regulation of IL-6. The clinical significance of CD147, IL-6, and three signaling molecules in CCA was emphasized by the demonstration of mRNA overexpression in CCA samples from public clinical datasets.

The differential expressions of CD147 in cancer cell lines were previously demonstrated [33]. The different expressions and degrees of glycosylation are related to the severity of diseases in the clinical samples. CD147 HG is highly expressed in malignant cancers [33]. The overexpression of CD147 was previously demonstrated in CCA samples [18], and higher expression of CD147, especially the HG isoform, promoted CCA cell invasion in vitro and in vivo [17]. Therefore, CD147 HG-overexpressing KKU-213A cells were selected for CD147 knockout generation. Reduced migration and invasion abilities were observed in CD147 KO clones [18], similar to previously described CD147 knockout hepatocellular carcinoma cells [33]. Membranous and soluble CD147s were identified in CCA cells. The functional significance of soluble CD147-promoting CCA invasion and IL-6 production in a CD147-dependent manner was observed in the current study. Roles of soluble CD147 on homophilic interaction were previously reported. It promotes the production of MMPs in cancer cells [22,27] and in fibroblasts [25,26,34], which later potentiated cancer invasion. It is worth noting that other CD147 ligands, e.g., cyclophilin A, might be responsible for IL-8 and GM-CSF inductions [30]. This requires further study to demonstrate that distinct CD147 ligands play diverse roles in CD147-dependent proinflammatory cytokine production.

CD147-potentiated CCA cell invasion is possible through the stimulation of cytokine production. This postulation was supported by the evidence that showed depletion of CD147 alleviated IL-6, IL-8, and GM-CSF production and cell invasion, and the invasion ability was rescued by the cytokine treatment in low CD147-expressing cells (KKU-055) and CD147-null cells (CD147 KO#1). Suppression of CD147 diminishing IL-1β, IL-6, and IL-8 production was previously demonstrated in head and neck cancer [16]. Actions of proinflammatory cytokines, such as IL-6, IL-8, and CCL2, on CCA cell migration and invasion were previously documented [10]. The production of GM-CSF was demonstrated in CCA cells, and it was proposed to promote neutrophil infiltration [35]. GM-CSF-inducing cancer cell invasion through induction of MMP production was established in colon cancer [36]. To the best of current knowledge, the relationship between CD147 and GM-CSF and the function of GM-CSF-promoting CCA invasion have never been reported in CCA.

The molecular mechanisms of CD147-mediated proinflammatory cytokine production were partially explored in this study using specific inhibitors. Akt, NF-κB, and p38 were selected from previously described CD147-related molecules [30]. The absence of CD147 reduced active forms, nuclear NF-κB, p-Akt, and p-p38, but not total forms of these proteins. To the best of our knowledge, CD147-mediated proinflammatory cytokine production, especially IL-6, via NF-κB, Akt, or p38, is described for the first time in this study. NF-κB seems to play a significant role in CD147-mediated proinflammatory cytokine production, while Akt and p38 demonstrated partial contributions. Decreased NF-κB nuclear translocation by DHMEQ significantly reduced all CD147-related proinflammatory cytokine production, whereas suppressions of Akt and p38 phosphorylations by specific inhibitors could reduce only IL-6, but not IL-8 and GM-CSF. The crucial roles of NF-κB-inducing inflammatory cytokine production were formerly demonstrated in inflammatory cells [37] and cancers [38]. The overexpression of NF-κB and inhibition of NF-κB nuclear translocation that suppresses CCA invasion were previously established [39,40]. The foremost contribution of IL-6 to CCA prognosis was documented [5]. The elevated plasma IL-6 increases risks of developing periductal fibrosis (a precancerous lesion) and cholangiocarcinoma to 19 and 150 times greater than a normal individual. The additional study regarding soluble CD147 and IL-6 in plasma might provide a benefit for future development of prognostic tools. The regulations of IL-6 production via the phosphatidylinositol 3-kinase (PI3K)-Akt pathway and p38 mitogen-activated protein kinase (MAPK) were demonstrated in colon cancer and myofibroblast cells [41,42]. The upregulation of IL-6 production is potentiated by specific proinflammatory cytokines, namely IL-1α, IL-1β, and TNFα. From the current results, however, the expressions of IL-1α, IL-1β, and TNFα were not altered in the CD147 KO clone (Figure 2A); hence, additional molecules are required.

Finally, to emphasize the significance of these findings in the clinical samples, the expressions of selected genes, including *BSG* (CD147), *IL-6*, *RELA* (p65), *AKT1* (Akt), and *MAPK14* (p38), in CCA tissues from three datasets were explored. The overexpression of the selected genes is generally observed. High CD147, IL-6, NF-κB p65, Akt, and p38 and their contribution to the worst disease progressions are previously reported [5,7,18,39,43]. The correlations between selected gene expressions, however, were not observed. These expressions might be due to the requirement of post-translational modification for protein activation rather than for gene regulations.

Altogether, the information suggests the contribution of CD147 to proinflammatory cytokine production, thus potentiating CCA invasion. The results of the current study are summarized in Figure 9. The relationship between loss of CD147 and decreased selected proinflammatory cytokines, namely IL-6, IL-8, and GM-CSF, is revealed. CD147-inducing IL-6 is dependent upon CD147 homophilic interaction. CD147-promoting IL-6, IL-8, and GM-CSF are involved in NF-kB nuclear translocation. Akt and p38 activations also participate in IL-6 induction, but not in IL-8 and GM-CSF production. Altogether, the information suggests the contribution of CD147 to proinflammatory cytokine production, thus potentiating CCA invasion. CD147 homophilic-interaction NF-kB nuclear translocation-dependent IL-6 production and CCA cell invasion are highlighted for the first time here. Thus, the current study suggests that the multifunction protein, CD147, is a promising target for CCA treatment. Suppression of CD147 or modulation of proinflammatory cytokines might benefit advanced CCA. Further studies on inflammation modulation and CCA suppression are mandatory.

## 4. Materials and Methods

### 4.1. Reagents and Antibodies

SB202190 (S7067) was purchased from Sigma-Aldrich (Darmstadt, Germany). MK-2206 was purchased from Santa Cruz Biotechnology (Santa Cruz, CA, USA). Dehydroxymethylepoxyquinomicin (DHMEQ) was kindly provided by Prof. Kazuo Umezawa, Aichi Medical University, Japan. Recombinant human (rh) EMMPIN/CD147 Fc chimera (NS0) protein (972-EMN), rhIL-6 (206-IL), rhIL -8 (208-IL), and rhGM-CSF (215-GM) were purchased from R&D system (Minneapolis, MN, USA).

The sources of antibodies were as follows; rabbit anti-NF-κB p65 (C-20, sc-372), polyclonal antibody (pAb), mouse anti-p-p38 (D8, sc-7973) monoclonal antibody (mAb), and mouse anti-p38 (A-12, sc-7972) mAb were from Santa Cruz Biotechnology; rabbit anti-p-Akt (S473, #9271) pAb, rabbit anti-Akt (#9272) pAb, horseradish peroxidase (HRP)-conjugated anti-rabbit IgG (#7074), and HRP-conjugated anti-mouse IgG (#7076) were from Cell Signaling Technology; anti-CD147 (MEM-M6/1) was from Abcam (Cambridge, UK); HRP-conjugated goat anti-mouse IgM was from Southern Biotech (Birmingham, AL, USA, #1020-05); anti-rabbit-IgG-FITC was from Invitrogen (Carlsbad, CA, USA, #656111). All cell culture-related reagents were from Gibco (Thermo Fisher Scientific, Waltham, MA, USA), and chemicals were purchased from Sigma-Aldrich. Otherwise, they will be specified.

### 4.2. Cell Culture

Three CCA cell lines, KKU-055, KKU-100, and KKU-213A, were established as previously described [44,45]. Cells were obtained from the Japanese Collection of Research Bioresources Cell Bank (JCRB Cell Bank, Osaka, Japan).

CD147 knockout KKU-213A cells (CD147 KO#1 and CD147 KO#2) were established as described elsewhere [18]. Briefly, gRNAs targeting exon 1 of CD147 were designed by CRISPRdirect software (http://crispr.dbcls.jp/, in March 2018); the sequences of gRNAs were 5′-CACCGCTGCTGGGATTCGCGCTGC-3′ and 5′-AAACGCAGCGCGAATCCCAGCAGC-3′. Oligonucleotides were ligated into a len-tiCRISPRv2 puro vector (Addgene, Teddington, UK) at the BsmB1 digestion site. CD147 gRNA-containing vector was co-transfected with pCMVR8.74 and pMD2.G vectors (Addgene) into 293T packaging cells using HilyMax transfection reagent (Dojindo Laboratories, Kumamoto, Japan). Conditioned media containing viral particles were added onto KKU-213A cells with 10 µg/mL polybrene (Sigma-Aldrich). CD147 knockout clones (CD147 KO) were selected by the flow cytometry technique using JSAN cell sorter (Bay Bioscience, Tokyo, Japan) and a limiting dilution technique. CD147 expression was monitored by Western blot analysis.

All cell lines were maintained in Dulbecco’s modified Eagle’s medium (DMEM) supplemented with 10% fetal bovine serum, 1% antibiotic-antimycotic in a humidified incubator at 37 °C with 5% CO_2_.

### 4.3. Conditioned Media (CM) Preparation, Total Protein Extraction, and Western Blot Analysis

To determine soluble CD147 in CCA-conditioned media, three CCA cell lines, namely KKU-055, KKU-100, and KKU-213A, were seeded and cultured for 24 h. After media were replaced, cells were cultured for another 48 h until they reached cell confluence. CM was centrifuged at 2000× *g* for 5 min at 4 °C to remove cell debris. CM was concentrated to reduce the volume to 1/50 time by Vivaspin^®^20 (GE Healthcare, Buckinghamshire, UK). The concentrated CM was immediately used for Western blotting.

Total protein preparation and Western blotting were performed as previously described [17]. Protein concentrations were determined using Bradford reagent (Bio-Rad Laboratories, CA, USA). The immunoreactivity was detected using the ECL^TM^ Prime Western Blotting Detection System (GE Healthcare, Buckinghamshire, UK). The signals were visualized by ImageQuant™ LAS4000 analyzer and ImageQuant TL software ((version 8.0), GE Healthcare, Uppsala, Sweden).

### 4.4. Human Inflammatory Cytokine Profile

To determine cytokines in CM, KKU-055, KKU-213A, and CD147 KO cells were seeded and cultured for 72 h to reach >90% confluence. Then, media were replaced, and cells were cultured for another 24 h. CM was collected and centrifuged at 2000× *g* for 5 min at 4 °C. CM was immediately used or stored at −80 °C until needed.

Cytokine profiles released from KKU-213A and CD147 KO#1 cells were determined using RayBio^®^ C-Series human cytokine antibody array C5 (#AAH-CYT-5, RayBiotech, Peachtree Corners, GA, USA) following the manufacturer’s recommendations. Briefly, membranes were blocked for 30 min at room temperature (RT), then incubated with CM at 4 °C overnight. After washing, the membranes were incubated with antibody cocktails for 2 h at RT, followed by three washes. Membranes were then incubated with horseradish peroxidase (HRP)-conjugated streptavidin for 2 h. The signals were measured using ImageQuant™ LAS4000 analyzer and ImageQuant TL software (GE Healthcare). The cytokine levels of CD147 KO cells were normalized with those of KKU-213A cells and presented as relative expressions of KKU-213A = 1.

### 4.5. Cytokine Determination by Enzyme-Linked Immunosorbent Assay (ELISA)

Selected cytokines were measured by ELISA. CCL-2, IL-6, IL-8, and GM-CSF secreted from KKU-055, KKU-213A, CD147 KO#1, and CD147 KO#2 cells were determined by Ready-Set-Go!^®^ human CCL-2 kits, Ready-Set-Go!^®^ human IL-6 kits, IL-8 human uncoated ELISA kits (eBioscience, San Diego, CA, USA), and human GM-CSF Quantikine ELISA kits (R&D system, Minneapolis, MN, USA) according to the manufacturer’s instructions. Absorption at 492 nm was determined by an ELISA plate reader (Sunrise Tecan GmbH, Salzburg, Austria).

### 4.6. Cytokine Treatment and Inhibitor Treatment

To determine the effects of cytokines on CCA cell invasion, three human cytokines, rhIL-6, rhIL-8, and rhGM-CSF, were used. KKU-055, KKU-213, or CD147 KO#1 cells were seeded into 6-well plates and cultured for 18 h. Then, cells were treated with 50 ng/mL rhIL-6, 100 ng/mL rhIL-8, or 25 ng/mL rhGM-CSF for 48 h. After cytokine removal, cells were tested for invasion ability.

To determine the significance of Akt, p38, NF-κB signaling pathways, MK-2206 (Akt phosphorylation inhibitor), SB202190 (p38 phosphorylation inhibitor), or DHMEQ (NF-κB nuclear translocation inhibitor) [46] was applied. KKU-213A cells were cultured in DMEM for 18 h and were treated with indicated concentrations of MK-2206, SB202190, or DHMEQ for 24 h. Later CM was collected for IL-6, IL-8, and GM-CSF determinations.

### 4.7. Cell Invasion by Boyden Chamber Assay

Cell invasion was analyzed using the Boyden chamber assay using the 8.0 μm pore insert (Corning Incorporated, New York, NY, USA) as previously described [17]. Briefly, CD147 KO#1 or KKU-055 cells were placed on the inserts in serum-free media, and serum-containing media was added into the lower chamber. CD147 KO clones and KKU-055 cells were cultured for 9 and 24 h. After that, cells were fixed, stained with crystal violet, and counted under a microscope using a 10X objective lens. At least five microscopic fields/inserts and three inserts were examined.

### 4.8. RNA Extraction and Reverse-Transcriptase-Polymerase Chain Reaction (RT-PCR)

Expressions of cytokine receptors were semiquantitatively measured by RT-PCR as previously described [40]. Total RNA was extracted by the TRIzol™ Reagent (Thermo Fisher Scientific), and cDNA was synthesized by Applied Biosystems™ High-Capacity cDNA Reverse Transcription Kit (Thermo Fisher Scientific) following the manufacturer’s protocol. RT-PCR was performed using the LightCycle 480^®^ real-time PCR system (Roche Diagnostic, Mannheim, Germany). Forty nanograms of cDNA, 0.4 μM primers, and LightCycle 480^®^ SYBR green I master mix (Roche Diagnostic) was included. The gene expression level was quantified by LightCycle 480^®^ Relative Quantification software (Roche Diagnostic). Glyceraldehyde 3-phosphate dehydrogenase (GAPDH) was an internal control. Oligonucleotide primers are listed in Table 1.

### 4.9. In Silico Study

Two sets of gene expression data, Accession IDs GSE26566 (PMID: 22178589) and GSE89749 (PMID: 28667006) were downloaded from the Gene Expression Omnibus (GEO) repository (https://www.ncbi.nlm.nih.gov/geo, in June 2020) using GEOquery package in R version 3.6.1 [47]. The expression data were quantile normalized with the lumi package and then log2 transformation [48]. The Illumina probes with the same gene symbols were combined using collapseRows [49]. Gene expression from TCGA-CHOL (CCA) was downloaded from The Cancer Genome Atlas (TCGA) database (https://www.cancer.gov/tcga, in June 2020) [50,51]. Using fragments per kilobase million (FPKM) normalization, log2 transformation of (FPKM+1) was demonstrated for TCGA data. The correlation matrix was performed with corrplot packages in R [52]. After hierarchical clustering and multidimensional scaling in R, sample outliers were excluded (data not shown). After the exclusion, there were 103 CCA and 6 normal intrahepatic bile duct samples from GSE26566, 115 CCA and 2 normal samples from GSE89749, and 36 CCA and 9 adjacent normal tissues from TCGA-CHOL.

### 4.10. Statistical Analysis and Graphic Creation

The data are presented as mean ± SEM from at least two independent experiments unless otherwise specified. The statistical differences between the control and experimental groups were determined by Student’s *t*-test. Statistical analyses were performed using the GraphPad Prism software version 8.0 (GraphPad Software, Inc., San Diego, CA, USA). A value of *p* < 0.05 was considered statistical significance.

The graphical abstract was created with BioRender (https://biorender.com/, in June 2020).

## Figures and Tables

**Figure 1 ijms-22-13496-f001:**
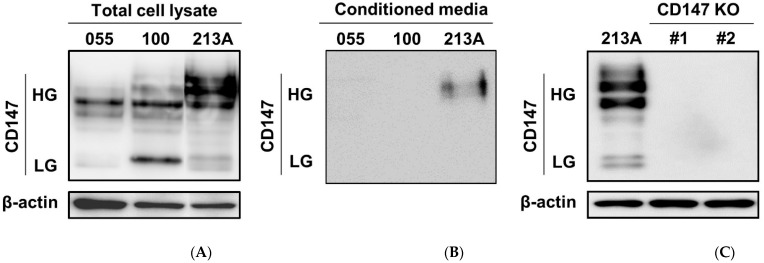
Expressions of CD147 in CCA and CD147 knockout (CD147 KO) cell lines. CD147 expressions were determined in total cell lysates (**A**) and conditioned media (**B**). (**C**) Total CD147s were compared between KKU-213A (213A) and CD147 knockout clones, CD147 KO#1 and CD147 KO#2. Highly glycosylated (HG) and lowly glycosylated (LG) CD147 were demonstrated. β-actin served as an internal control. 055; KKU-055, 100; KKU-100.

**Figure 2 ijms-22-13496-f002:**
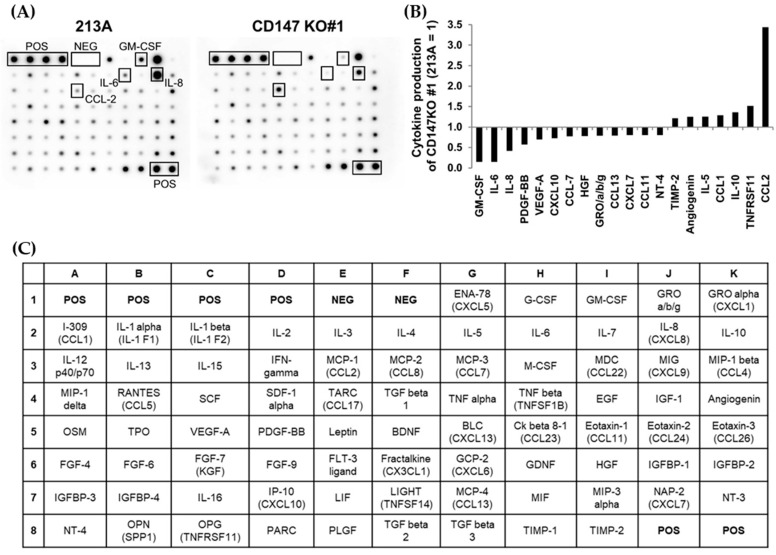
Proinflammatory cytokine production in parental cells, KKU-213A (213A), and CD147 knockout clone #1 (CD147 KO#1). (**A**) Proinflammatory cytokines in KKU-213A and CD147 KO#1 conditioned media (CM) were determined by human cytokine arrays. (**B**) Relative levels of twenty differentially expressed cytokines. Levels in parental cells were set as 1. (**C**) A map of the human cytokine array used in the experiment. POS = positive control spots, NEG = negative control spots.

**Figure 3 ijms-22-13496-f003:**
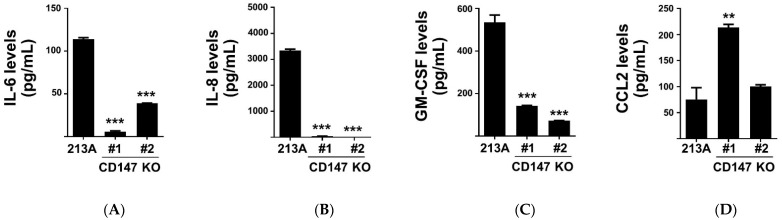
The selected four inflammatory cytokine levels in KKU-213A (213A) and CD147 KO clones (#1 and #2) CM. (**A**) IL-6, (**B**) IL-8, (**C**) GM-CSF, and (**D**) CCL2 levels are shown. The data are means ± SEM from representative experiments. ** *p* < 0.01, *** *p* < 0.001.

**Figure 4 ijms-22-13496-f004:**
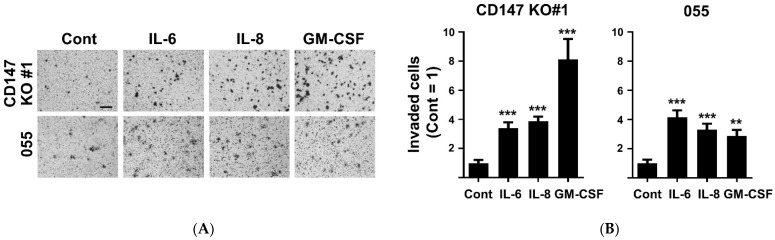
IL-6, IL-8, and GM-CSF potentiated CCA invasion independently to CD147 expression. (**A**) Cytokine-stimulated invasion in KKU-055 (055) and CD147 KO#1 cells were determined by the Matrigel-coated Boyden chamber assay. (**B**) Relative cell number invasions compared between unstimulated (Cont.) and IL-6-, IL-8-, and GM-CSF-treated cells. Mean invaded cell number in unstimulated condition was set as 1. The data are means ± SEM from representative experiments. ** *p* < 0.01, *** *p* < 0.001. Bar = 50 μm.

**Figure 5 ijms-22-13496-f005:**
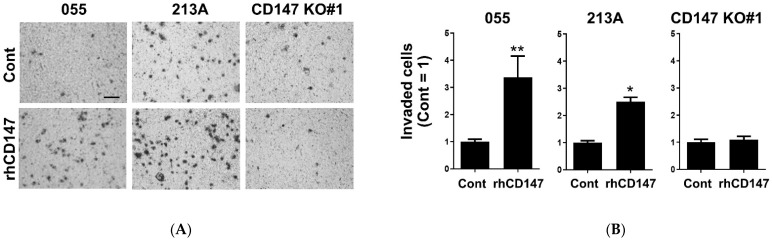
Recombinant hCD147 promoted CCA invasion in CD147-expressing cells. (**A**) CD147-mediated CCA invasion determined by Matrigel-coated Boyden chamber assay. (**B**) Relative cell invasion in unstimulated (Cont) and rhCD147-treated KKU-055 (055), KKU-213A (213A), and CD147 KO#1 cells. Mean invaded cell numbers in unstimulated conditions were set as 1. The data are means ± SEM from representative experiments. * *p* < 0.05, ** *p* < 0.01. Bar = 50 μm.

**Figure 6 ijms-22-13496-f006:**
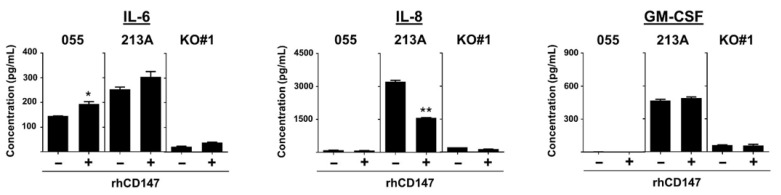
Recombinant hCD147-potentiated IL-6 production in CCA cells. (**A**) IL-6, (**B**) IL-8, and (**C**) GM-CSF were determined in rhCD147-treated KKU-055 (055), KKU-213A (213A), and CD147 KO#1 (KO#1) cells. Cytokine levels in CM were measured by ELISA. The data are means ± SEM from representative experiments. * *p* < 0.05, ** *p* < 0.01. − = untreated, + = rhCD147-treated condition.

**Figure 7 ijms-22-13496-f007:**
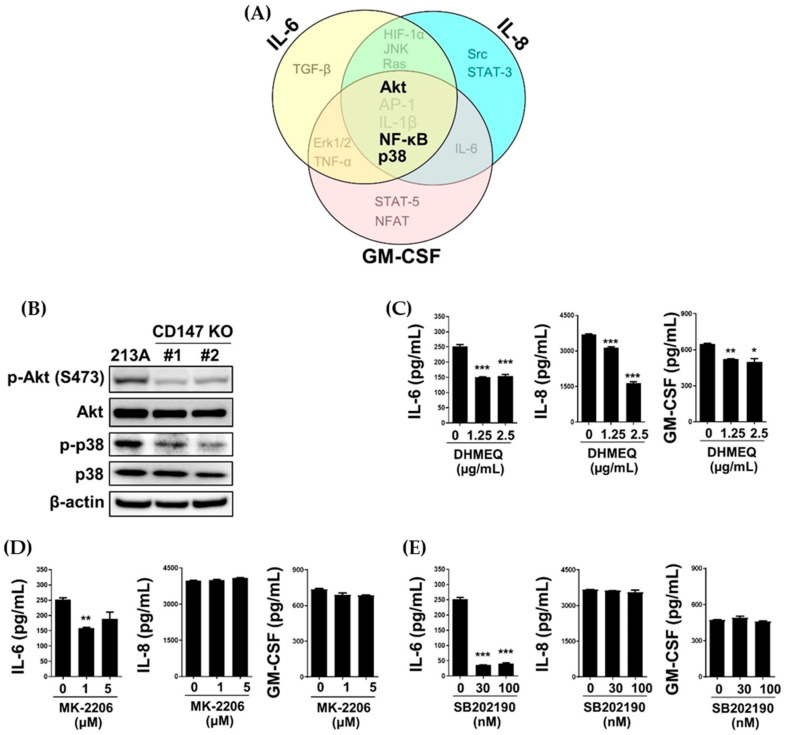
Potential molecules involved in CD147-induced proinflammatory cytokine production. (**A**) Regulatory molecules involved in selected proinflammatory cytokine production are grouped. The potential molecules that are controlled by CD147 and regulate IL-6, IL-8, and GM-CSF are emphasized by bold letters. (**B**) The expressions of p-Akt, Akt, p-p38, and p38 in parental and CD147 KO clones (#1 and #2). IL-6, IL-8, and GM-CSF levels were determined in (**C**) DHMEQ, (**D**) MK-2206, and (**E**) SB202190-treated KKU-213A cells. The data are means ± SEM from representative experiments. The numbers on the *x*-axis indicate the inhibitor concentrations. * *p* < 0.05, ** *p* < 0.01, *** *p* < 0.001.

**Figure 8 ijms-22-13496-f008:**
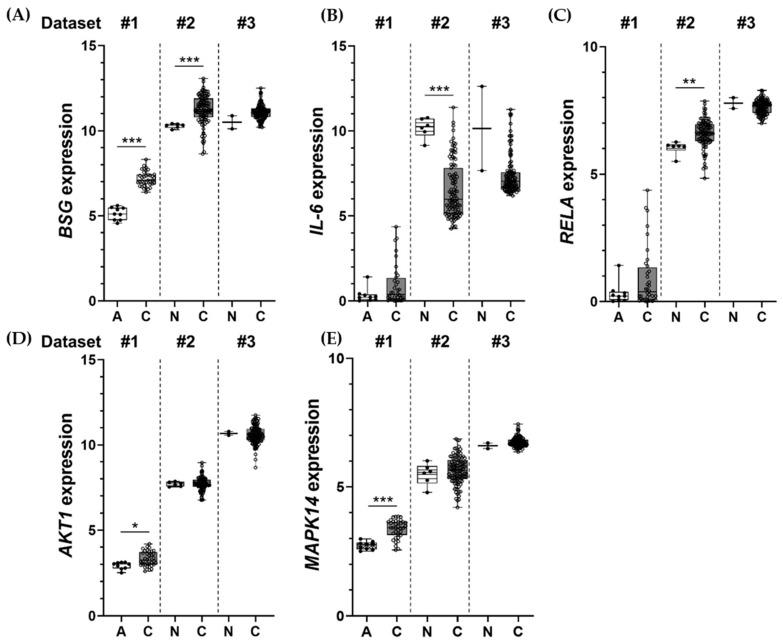
Expression profiles of BSG (**A**), IL-6 (**B**), RELA (**C**), AKT1 (**D**), and MAPK14 (**E**) in CCA tissues (C) compared to adjacent noncancerous tissues (A) or normal bile ducts (N) from 3 different datasets: #1 = TCGA-CHOL, #2 = GSE26566, and #3 = GSE89749. Normalized expression levels of individual samples are presented with individual dots; #1 = log_2_(FPKM+1), and #2 and #3 = log_2_ (quantile normalization). Dots represent an expression level from an individual sample. Bars indicate minimum to maximum levels. * *p* < 0.05, ** *p* < 0.01, *** *p* < 0.001.

**Figure 9 ijms-22-13496-f009:**
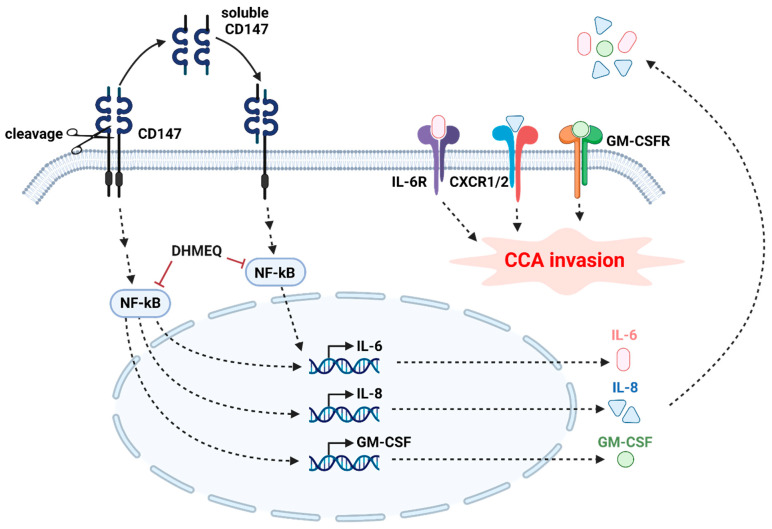
Proposed mechanism of CD147-induced cytokine production and mediated invasion in CCA. Roles of soluble CD147 are demonstrated. The involvements of NF-κB translocation and CD147 homophilic interaction in the CD147-dependent mechanism are suggested. Modulation of CD147, NF-κB translocation, or proinflammatory cytokine actions might, therefore, slow the disease progression and improve the prognosis.

**Table 1 ijms-22-13496-t001:** List of primers for RT-PCR.

Genes	Forward Primers (5′→3′)	Reverse Primers (5′→3′)
*IL-6R*	CATTGCCATTGTTCTGAGGTTC	GTGCCACCCAGCCAGCTATC
*CXCR1*	CCTTCTTCCTTTTCCGCCAG	AAGTGTAGGAGGTAACACGATG
*CXCR2*	TTGGCTCTTCTTCAGGGCACACTT	CAGGGGCACAATCTCGGCTCAC
*GM-CSFR*	GCAGACGTCCGCATCTTGA	CCGTCGTCAGAACCAAATTCA

## Data Availability

The data presented in this study are available on request from the corresponding author. The data are not publicly available due to privacy.

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
