# Peer review of "Homophilic Interaction of CD147 Promotes IL-6-Mediated Cholangiocarcinoma Invasion via the NF-κB-Dependent Pathway"

_ijms, 2021, doi:10.3390/ijms222413496_

Round 1

Reviewer 1 Report

This study explored the role of CD147 in CCA using cell lines and knock-out models. Knock-out of CD147 reduced IL-6, IL-8 and GM-CSF secretion. This was confirmed by ELISA and also using another knock-out line. When IL-6, IL-8 and GM-CSF were applied to CD147 knock-out or CD147 low expressing cell lines, they induced cell invasion. These results are promising however in the next figures; there are several points that penalise the conclusion.

  1. In figure 5, only one knock-out line was tested. This result is important for the statement regarding homophilic interaction of CD147. Thus, another knock-out line needs to be included.
  2. Figure 6, only one knock-out line was used as well. Importantly, there seems to be an increase in IL-6 level in the knock out line. This again needs to be checked in another line.
  3. In figure 6, levels of IL-8 and GM-CSF were not increased with rhCD147 treatment. This did not support their hypothesis regarding CD147 dependent up-regulation of these two cytokines.
  4. To confirm the role of NF-kB, there should be another experiment where cells are treated with hCD147 and DHMEQ simultaneously. Up-regulation of IL-6 should be then diminished upon inhibition of NF-kB.
  5. The authors have only used recombinant CD147, but can CD147 naturally secreted from those cells act the same way? In another word, can conditioned medium from CD147 high expressing cells stimulate IL6 production in CD147 low expressing cells compared to conditioned medium from CD147 knock-out?
  6. Please describe briefly regarding the knock-out model employed here; some information that are necessary for the readers to follow the experiment without having to check the reference such as what is the parental cell, the knock-out system used.

Reviewer 2 Report

The manuscript of Dana et al reports a role of the CD147 transmembrane protein in invasiveness of cholangiocarcinoma cells and in production of proinflammatory cytokines.

First of all, this manuscript is really carefully written and presents a nice study. All conclusions are based on experimental data. Materials and methods section gives all nessesary information for reproducibility. The text contains no grammatical errors or mistypos, so the authors did a good job.

Actually, my only comment concerns study design. It is well known that CD147 is not just a protein but a cofactor of MCT pyruvate/lactate transporters. So the first question is whether differences in glycolysis were evaluated for the presented CCA cell lines. The second question is about possible impact of inhibitors of glycolysis/lactate dehydrogenase/CD147 itself on cell invasiveness and cytokine production was tested and why. And the third question concerns choice of medium for such studies: human blood plasma contains app 1 mM lactate, and its level is increased in tumor surrounding. However, this and many other similar studies are performed in the settings of standard media such as DMEM supplemented with artificially high level of pyruvate. So will the results remain the same if DMEM is substituted with lactate-containing medium? This is especially important since the protein that controls its flux is discussed.

Nevertheless, the presented study is good in its form as well.

Round 2

Reviewer 1 Report

I would like to thank the authors for spending time to provide all the answers and more data. The new data, added to supplementary section, provides some more proof and due to the lack of time provided for revision, the manuscript can be published in the present form.

Author Response

Dear Reviewer,

 Thank you very much for your valuable comments.

 With best regards,

 Kulthida